# Toxicity of Beauty Salon Effluents Contaminated with Hair Dye on Aquatic Organisms

**DOI:** 10.3390/toxics11110911

**Published:** 2023-11-07

**Authors:** Letícia C. Gonçalves, Matheus M. Roberto, Paloma V. L. Peixoto, Cristina Viriato, Adriana F. C. da Silva, Valdenilson J. A. de Oliveira, Mariza C. C. Nardi, Lilian C. Pereira, Dejanira de F. de Angelis, Maria A. Marin-Morales

**Affiliations:** 1Department of General and Applied Biology, Institute of Biosciences, São Paulo State University (Unesp), Av. 24-A, 1515, Bela Vista, Rio Claro 13506-900, SP, Brazil; lcgoncalvess@gmail.com (L.C.G.); dricanana7@gmail.com (A.F.C.d.S.); zittos97@gmail.com (V.J.A.d.O.); dejanira.angelis@unesp.br (D.d.F.d.A.); 2University Center of Hermínio Ometto Foundation (FHO), Av. Dr. Maximiliano Baruto, 500, Jardim Universitário, Araras 13607-339, SP, Brazil; mariza@fho.edu.br; 3Center for Evaluation of Environmental Impact on Human Health (TOXICAM), Botucatu Medical School, São Paulo State University (Unesp), Av. Prof. Mário Rubens Guimarães Montenegro, s/n, Rubião Júnior, Botucatu 18618-687, SP, Brazil; paloma.peixoto@unesp.br (P.V.L.P.); cristina.viriato@unesp.br (C.V.); lilian.pereira@unesp.br (L.C.P.); 4Department of Pathology, Botucatu Medical School, São Paulo State University (Unesp), Av. Prof. Mário Rubens Guimarães Montenegro, s/n, Rubião Júnior, Botucatu 18618-687, SP, Brazil; 5Department of Bioprocesses and Biotechnology, São Paulo State University (Unesp), R. Dr. José Barbosa de Barros, 1780, Fazenda Experimental Lageado, Botucatu 18610-307, SP, Brazil; 6School of Agriculture (FCA), São Paulo State University (Unesp), Av. Universitária, 3780, Fazenda Experimental Lageado, Botucatu 18610-034, SP, Brazil

**Keywords:** ecotoxicity, aquatic ecotoxicology, emerging environmental contaminants, cosmetic residues, zebrafish embryotoxicity, lethality test

## Abstract

Cosmetic residues have been found in water resources, especially trace elements of precursors, couplers, and pigments of hair dyes, which are indiscriminately disposed of in the sewage system. These contaminants are persistent, bioactive, and bioaccumulative, and may pose risks to living beings. Thus, the present study assessed the ecotoxicity of two types of effluents generated in beauty salons after the hair dyeing process. The toxicity of effluent derived from capillary washing with water, shampoo, and conditioner (complete effluent—CE) and effluent not associated with these products (dye effluent—DE) was evaluated by tests carried out with the aquatic organisms *Artemia salina*, *Daphnia similis*, and *Danio rerio*. The bioindicators were exposed to pure samples and different dilutions of both effluents. The results showed toxicity in *D. similis* (CE_50_ of 3.43% and 0.54% for CE and DE, respectively); *A. salina* (LC_50_ 8.327% and 3.874% for CE and DE, respectively); and *D. rerio* (LC_50_ of 4.25–4.59% and 7.33–8.18% for CE and DE, respectively). Given these results, we can infer that hair dyes, even at low concentrations, have a high toxic potential for aquatic biota, as they induced deleterious effects in all tested bioindicators.

## 1. Introduction

The aquatic environment is highly complex, characterized by diverse ecosystems such as rivers, lakes, estuaries, seas, and oceans [1]. However, this classification is simplistic, because these ecosystems are dynamic structures with specific characteristics [2]. When pollutants are released into the environment, they can disperse inside a distinct compartment, such as soil, water, and air, and be transported among them. Despite fabrication processes that generate distinct effluents, generally, cosmetics are released as domestic sewage, bypass the conventional wastewater treatment system (when present), and reach different receiving waters, such as freshwater bodies, and then marine environments [3]. During the movement of a contaminant, several interactions between abiotic and biotic factors occur, reaching exposed organisms. Once it enters into the biota, bioaccumulation becomes an impairment factor for living beings, and through the trophic chain, biomagnification worsens the scenario [4,5].

The integrity of aquatic ecosystems has been disturbed by anthropogenic activities, derived from changes in lifestyle, social context, agriculture, industry, and technological advances [6]. The changes in these environments happen directly or indirectly, and their ecotoxicological damage is unpredictable and of high complexity [7]. Hüttl et al. [7] also highlight that the maintenance of environmental quality will only be possible if interdisciplinary studies are implemented, which include a multifaceted view of the complexity of the environmental problem.

Regarding water sustainability, there is also a concern about establishing a water ethics program, with proposals for the conduct of rigid decisions, especially for natural environments that are little or not fully known [7]. This call for a new water ethics policy is also based on concerns about the increased degradation of water systems [7,8], To attend to the concept of “one health” (the decompartmentalization of human, animal, and ecosystem health) and reveal unknown impacts, the interlocution among different study areas should be strengthened, such as environmental chemistry, human toxicology, and ecotoxicology [9].

In recent years, some studies of aquatic ecosystems have shown the presence of traces of dyes, precursors, and couplers, whose origin would be from the process of hair dyeing [10]. The effluents generated in beauty salons are, in most cases, directly disposed of into the environment or incorporated into the urban sewer system. The characteristics of this effluent are much closer to industrial than to domestic effluent [5]. However, as the conventional system used in sewage treatment plants is not completely efficient for the removal of this pollutant, many cosmetic products remain in surface waters [5,11].

As an example, the work of Yurtsever [12] describes how glitter particles from some cosmetics are found in the environment, first in sewage sludge, then in rivers (water and sediment), and finally in oceans. Hair dyes also constitute a serious environmental threat, and their effluents must be investigated.

The indiscriminate disposal of toxic agents in the environment can be characterized as a risk factor for the survival of organisms [13,14], in a more intensive way for aquatic species whose entire life cycles can be assumed to be aqueous [15]. Considering the current high consumption of hair dyes and the dyeing process, beauty salons generate a large volume of complex effluents (with compounds belonging to groups of preservatives, humectants, organic solvents, surfactant oils, and also with dyes based on heavy metals, such as lead, cadmium, chromium, and arsenic) [16,17].

Studies of water quality and the polluting load of effluents are usually done by physicochemical analyses [18]. However, this type of evaluation is very limited, as the analyses are insufficient to estimate the real toxic potential of a given contaminant [19,20]. On the other hand, ecotoxicological studies allow the meticulous assessment of the toxicity of both effluents and their receiving bodies. These tests are performed with aquatic bioindicators of high sensitivity, whose patterns of effects may reflect the possible damages on other biological categories [21]. Bioindicators are tools widely used by environmental regulatory bodies [18] to identify effects on the aquatic biota of inland, estuarine, and marine waters, and can be used both in laboratory and field conditions [22].

With these tests, it is also possible to establish permissible limits for different substances, as well as to evaluate the impact of complex mixtures on the aquatic organisms of the receiving bodies [19]. According to Domingues and Bertoletti [23], when conducting ecotoxicological tests on aquatic organisms, it is important to use taxonomic groups that are representative of the ecosystem. Also, it is recommended to use organisms from different trophic levels to get an idea of the natural sensitivity variability of organisms in the ecosystem under consideration.

Given the health and environmental problems related to hair dyes, it is essential to conduct studies that help in understanding the interaction of these compounds with the aquatic biota, as well as the possible damage that these contaminants may induce in living beings in general. In the present study, by in vivo tests, the toxicity and embryotoxicity of two types of effluents generated in the hair dyeing process were evaluated on three bioindicators of various trophic levels: *Artemia salina*, *Daphnia similis*, and *Danio rerio*.

## 2. Materials and Methods

### 2.1. Obtaining, Storing, and Determining Effluent Samples

To perform the tests, effluents generated from the hair dyeing process were collected from a beauty salon (city of Rio Claro, São Paulo, Brazil). Two types of effluents were collected under the authorization of the owner of the establishment and its customers: (1) effluent derived from the freshly dyed hair washed using water, shampoo, and conditioner (complete effluent—CE); and (2) effluent composed of wash water and only the dye removed from the hair (dye effluent—DE), i.e., without shampoo and conditioner. Brown hair dye was applied to the hair according to the manufacturer’s instructions, for 40 min. Then, a complete wash was done (removal of excessive dye by washing with water, shampoo, and conditioner), generating CE. Following the same protocol of the dyeing process on another person, the washing was performed only with tap water (until the complete removal of the dye that was not fixed to the hair), generating DE. To remove possible impurities, such as hair strands, the samples were filtered by qualitative filter paper (80 g/m^2^) and stored, separately, in properly identified polypropylene carboys (20 L capacity—much more effluent was discarded). To avoid the degradation of the components of the effluents until the assembly of the tests, samples were stored in a freezer (−10 °C).

### 2.2. Treatments

The bioassays were performed with both types of effluents (DE and CE), followed by their respective control tests. The bioassays performed with *A. salina* and *D. rerio* were developed with the same concentrations/dilutions of DE and CE, as follows: pure sample (100%) and dilutions of 3.125, 6.25, 12.50, 25.00, and 50.00%. The bioassays with *D. similis* were performed with pure sample (100%) and dilutions of the effluents (1.00, 3.00, 3.125, 6.25, 12.50, 25.00, and 50.00%). For DE and CE EC_50_ determination, 0.5 and 5.0% treatments were added, respectively, for test validation.

### 2.3. Bioassays

#### 2.3.1. Test Method with *Artemia salina*

The bioassays of acute ecotoxicity with *Artemia salina* followed the guideline of ABNT NBR 16530/2016 [24]. For greater reliability of the results obtained in the test, larvae (nauplii) were obtained from the hatching of selected and vacuum-packed cysts (*Artemia salina* RN^®^, Natal, Brazil) acquired in an aquarium store. The cysts and nauplii were maintained in optimal conditions of hatching and growth (25.0 ± 2.0 °C, salinity around 3.2%, and slightly alkaline pH ~8.0).

The experiment was carried out in a glass aquarium of 19 × 10 × 15 cm dimensions, with one of the walls covered by aluminum foil. The volume of the aquarium was separated into two parts by a partition plate, uniformly perforated (~2 mm in diameter, spaced by 4 mm). The aquarium received 500 mL of reconstituted water (synthetic seawater), prepared by dissolving commercial sea salt (Blue Treasure Reef Sea^®^, Qingdao Sea-Salt Aquarium Technology Co., Ltd., Qingdao, China) in distilled water, according to the manufacturer’s guidelines.

The aquarium was accommodated in a biological oxygen demand (BOD) incubator with the temperature controlled at 25.0 ± 2.0 °C, constant aeration, and lighting. One of the parts of the aquarium received constant illumination (aquarium wall without aluminum foil cover), i.e., it was exposed to a fluorescent lamp of 15 W. The other part, covered with aluminum foil, and therefore without lighting (dark), received 0.3 g/L of *A. salina* cysts. The light/dark system of the aquarium was prepared to attract the newly hatched larvae to the light side, forcing them to cross the partition (through the holes) due to the positive phototropism they have, standardizing the organisms used in the test. Incubation was carried out for a period of 48 h.

After the incubation period, the *A. salina* nauplii were separated and exposed to different effluent dilutions (described in Section 2.2) for 48 h in a static system. Test tubes received 10 individuals and 10 mL of sample, and each treatment was organized in quadruplicate. Through the lethality results, the LC_50_ of the samples of each effluent was determined, characterizing the concentration responsible for causing death in 50% of the organisms.

The exposure of the organisms was accompanied by a negative control (NC), performed only with reconstituted water. In addition, the sensitivity of the bioindicator was ensured by exposure to a reference substance, sodium dodecyl sulfate (SDS—0.625, 1.25, 2.5, 5.0, 10.0, and 20.0 mg/L), solubilized in reconstituted water.

#### 2.3.2. Test Method with *Daphnia similis*

The acute toxicity bioassays carried out with the microcrustacean *Daphnia similis* followed the guideline ABNT NBR 12713/2016 [25]. The individuals of *D. similis* used in the experiment were obtained from cultures maintained in the Water Toxicity Laboratory, from the Department of General and Applied Biology, São Paulo State University (UNESP), Rio Claro, Brazil. The cultures were maintained following the technical standards recommended by ABNT NBR 12713/2016 [25].

To assess the toxicity of the effluents, a total of five neonates, aged between 6 and 24 h, were exposed to 10 mL of the samples in treatments organized in quadruplicate. Different concentrations of the samples were evaluated (according to Section 2.2), accompanied by a negative control (NC), consisting only of diluted water (culture water of the organisms—hardness 40 mg/L to 48 mg/L and pH in the range of 7.2 to 7.6) [25]. The test tubes containing the organisms were kept in a BOD incubator at 20 ± 2 °C, in the dark, without feeding, for 48 h. At the end of the exposure period, the toxicity of the effluents was evaluated by the number of mobile and immobile organisms (lethality parameter).

#### 2.3.3. FET Test

##### Zebrafish Maintenance and Spawning

The procedures for the care and use of animals were approved by the Ethiculture, and spawning and test procedures were conducted according to OECD guidelines (OECD No. 236, Commission, 2013) [26], in the Zebrafish Animal Laboratory located at UNIPEX—Experimental Research Unit of Botucatu Medical School, UNESP, Brazil, approved by CEUA n° 1392/2021.

Adult male and female zebrafish (*Danio rerio*) were kept separately in a recirculating aquaculture system containing mechanical biological filtration and disinfection. The temperature was 26.0 ± 1.0 °C in a 14 h light/10 h dark cycle. Water hardness was approximately 150 mg CaCO_3_/L and pH was 6.8 to 8.5. The dissolved oxygen concentration was above 6.0 mg/L. Fish were fed with *Artemia salina* three times daily. Sexually mature male and female zebrafish were transferred into a spawning tank separated by a partition and placed in a dark and quiet environment (2 males:1 female), and breeding began with the presence of light the following day. Approximately 35 min after spawning, fertilized eggs were collected and selected with an inverted microscope (Motic^®^, AE2000). Viable eggs were selected for the embryotoxicity test.

##### Fish Embryo Exposure Assays

The Fish Embryo Acute Toxicity (FET) test (OECD No. 236, Commission, 2013) [26] was performed in independent replicates with 24-well plates (SPL Life Sciences), one embryo per well. The experimental design consisted of four treatments with different dilutions for each effluent (DE and CE): 3.125, 6.25, and 12.5%; a negative control (NC) (reconstituted water—ISO-7346); and a positive control (4.0 mg/L 3,4-dichloroaniline). In the NC, 24 embryos were exposed to the ISO-7346 water. On the other plates, 20 eggs were exposed to each different dilution for each effluent (DE and CE), and four eggs in ISO-7346 water were the internal NC. For each effluent, three replicates were used (*n* = 3).

Embryonic development was evaluated at 8, 24, 48, 72, 96, 120, and 144 h post-fertilization (hpf) with an inverted microscope (AE2000, Motic^®^, Barcelona, Spain). Mortality was recorded daily according to lethality observations: coagulation of embryos, lack of somite formation, non-detachment of the tail, and lack of heartbeat. Microphotographs (24 and 144 hpf) were recorded with an Olympus MVX10 magnifier. Images were processed with Fiji-ImageJ software (National Institutes of Health, Bethesda, MD, USA). The LC_50_ (median lethal concentration—for embryo-lethality) was determined at 24, 96, and 144 hpf for DE and CE, and it was determined whether there was a significant difference in these values between the effluents. For DE LC_50_ determination, a 0.5% treatment was added for test validation.

### 2.4. Physicochemical Analysis

Water samples were measured after collection using the Horiba multiparameter probe (Multi Water Quality Checker U-50 Series, Kyoto, Japan). The parameters evaluated by the probe were: temperature, pH, conductivity, turbidity, dissolved oxygen, total dissolved solids, and salinity.

### 2.5. Spectrophotometric Analysis

The evaluation of the remaining concentration of hair dye in the effluents studied (DE and CE) was performed using UV-visible spectrophotometry (UV-5100, Global Trade Technology^®^, Jaboticabal, Brazil). For this, a standard solution was prepared with the commercial mixture at 12.0 g/L, the same used in the hair coloring process to obtain the effluents. Then, the absorbance reading was performed by spectrophotometry, using the external calibration method, to identify the wavelength of maximum light absorption by the mixture. The wavelength corresponding to maximum light absorption occurred at 290 nm. Thus, absorbance readings of the standard solution and different dilutions of the same mixture (0.6, 1.2, 2.4, 4.8, 6.0, and 12.0 g/L) were performed to obtain the standard curve by linear regression. After making the standard curve, an aliquot of each effluent was subjected to centrifugation for 4 min at 484× *g* (Centrifuge 80-2B, Centribio^®^, São Paulo, Brazil), to remove possible impurities, and read at 290 nm. Thus, based on the absorbances obtained, the concentration of the capillary dye mixture of each effluent was determined using the equation of linear regression [27,28,29].

### 2.6. Statistical Analysis

For *D. rerio* data, the median lethal concentrations of the effluents DE and CE were estimated using the GW-Basic 3.0 Software, according to the “Trimmed Spearman Karber” statistical method [30]. Differences in LC_50_ values were determined using Student’s *t*-test. The data obtained in this study were submitted for analysis of variance (one-way ANOVA). Shapiro-Wilk and Levene tests respectively verified the subjects of normality and homogeneity of the variances of the residues. When significant differences between treatments were detected (*p* < 0.05), Tukey’s a posteriori test was used. In the case of non-parametric data, the Kruskal-Wallis test was used, followed by the multiple comparison test as an a posteriori test [31].

The determinations of LC_50_ (*A. salina*) and EC_50_ (*D. similis*) were performed by the “Trimmed Spearman Karber” method, as described in ABNT NBR 16530/2016 and ABNT NBR 12713/2016, respectively [24,25].

## 3. Results

### 3.1. Artemia salina

The mortality of the bioindicator *A. salina* for the different dilutions of DE and CE is shown in Table 1. The results indicate mortality of 100% for concentrations of 25, 50 and 100% for both effluents and higher mean percentages for non-lethal DE concentrations. The estimated mean values for LC_50_ were statistically determined as 3.874% for DE and 8.327% for CE.

### 3.2. Daphnia similis

The mean mortality of *D. similis* exposed to the different dilutions of DE and CE effluents are shown in Table 2. The results indicate mortality of 100% for concentrations of 3.00, 3.125, 6.25, 12.50, 50.00, and 100.00% of the DE. The treatment performed with CE also induced mortality of 100% for the same concentrations, except for 3.00%. The mean EC_50_ values were statistically determined as 0.54% for DE and 3.43% for CE.

### 3.3. FET Test

The LC_50_ for 24 hpf for zebrafish (*Danio rerio*) embryos exposed to DE was 7.33% (95% confidence interval: 6.25–8.65%) and to CE was 4.59% (95% confidence interval: 3.94–5.36%). At the end of the 144-h post-fertilization exposure, the LC_50_ was 8.18% (95% confidence interval: 7.13–9.4%) for DE and 4.25% (95% confidence interval: 3.67–4.93%,) for CE. The results of LC_50_ showed significant differences for all times of exposure (24, 96, and 144 hpf) between DE and CE (Table 3). Embryonic coagulation was the most observed lethal endpoint in DE and CE treatments (Figure 1). High mortality was verified at 24 hpf in the effluents at concentrations ≥6.25% (Figure 1a,b; examples of mortality are shown in Figure 1c). The data obtained showed that CE exposure is more toxic than exposure to DE for zebrafish embryo-larval development.

The survivor embryos in CE treatments showed no malformations significantly up to 144 hpf (Figure 1d), but some results showed pericardial and yolk sac edema in embryos of DE treatments (Figure 2). The highest percentage of malformations was present at 48 h. After 120 h, the edema disappeared, showing an adaptive response of the embryo.

### 3.4. Physicochemical Analysis

The results of the physicochemical analyses of the DE and CE samples, obtained with the multiparameter probe (Table 4), showed differences in turbidity, for CE, and in dissolved oxygen and turbidity for DE, when compared with two Brazilian laws established by the National Environment Council (Conselho Nacional do Meio Ambiente—CONAMA—n° 357/2005 e n° 430/2011) [32,33] and an international document from the European Union concerning urban wastewater treatment (Council Directive nº 91/271/EEC of 21 May 1991; EUR-LEX, 1991) [34].

### 3.5. Spectrophotometric Analysis

The absorbance measured for DE was 0.192, and that for CE was 0.342. Thus, the concentration of the hair dye mixture was determined by the analytical curve (Figure 3), resulting in values of 1.505 g/L for CE and 0.895 g/L for DE.

## 4. Discussion

The assessment of the polluting loads of the studied effluents (DE and CE) was performed by physicochemical characterization, measured by a multiparameter probe. In this analysis, the parameters of pH, temperature, conductivity, dissolved oxygen, total soluble solids, turbidity, and salinity were considered. These parameters obeyed Resolution No. 357/2005 of CONAMA, which classifies Brazilian freshwater bodies into different categories, according to the water quality standards [32].

Another legislation used was Resolution No. 430/2011 of this same Brazilian council, which provides conditions, parameters, and standards for the discharge of effluents into water bodies [33]. According to the few reference parameters of this resolution, it is noted that the DE and CE presented temperature, pH, and salinity within the acceptable range. In addition to these Brazilian legislations used to classify the effluents studied here, the European Union Council Directive No. 91/271/EEC [34] was also considered to evaluate the quality of the effluents, regarding the requirements recommended at the international level.

The pH measured for both effluents (6.19 and 5.14 for DE and CE, respectively) is within the standards defined by Resolution 357/2005 [32] and by Resolution 430/2011 [33], which determines a pH range for effluents between 5.0 and 9.0. The determination of this pH range is due to the water quality safety policy because values below 5.0 or above 9.0 can cause a decrease in cellular activity, making survival impossible for aquatic organisms [32,33]. The pH is considered an important variable of water quality, as it influences several biological and chemical processes. Changes in pH can affect the survival of aquatic organisms as they have pH-dependent metabolic activities [35]. According to Omer [36], most aquatic organisms are already adapted to a specific pH and changes can impair their survival; a pH below 3.0 is commonly fatal. Also, this author highlights that pH affects the solubility of other chemicals in water, which endangers the exposed biota. The maintenance of the recommended values for this parameter is essential, as preserving the water quality also directly meets two of the goals established by the UN 2030 Agenda (goals 6 and 14) [37] and the water ethics policy [7,8].

According to the Annual Inland Water Quality Report in the State of São Paulo [38], Brazil, conductivity can be considered the numerical expression of the capacity of water to conduct electric current. That is, considering a freshwater environment, the conductivity value expresses the quantity of salt in the water, representing an indirect measure of pollutant concentration. Conductivity also acts as an indication of changes in water bodies, because this parameter tends to be constant when there is no anthropic interference in the environment [39].

Although there is no legislation that defines effluent conductivity limits, impacted environments are those that have conductivity levels higher than 100 μS/cm [40]. The DE and CE samples presented conductivity values higher than 100 μS/cm (391.0 and 204.0 μS/cm, respectively). Therefore, because this parameter is considered an indirect measure of the concentration of pollutants [40], it can be inferred that effluents contaminated with hair dye can compromise the water quality of their receiving water bodies.

Rodrigues et al. [41] suggest that the amounts of dye that are fixed in the hair may vary according to the type of dyed hair, a fact that could generate variation in the composition of the effluents of the hair washes. Considering this concern, in the present study a spectrophotometric analysis was performed to quantify the presence of brown hair dye in each of the effluents evaluated. In this analysis, a higher concentration of dye was observed in the CE (1.505 g/L) than in the DE (0.895 g/L), which can be explained by the use of shampoo during hair washing. Professional shampoos have ingredients in extremely concentrated forms or special anionic or cationic detergents, capable of removing the residues of chemicals applied in the hair coloring process [42].

Another physicochemical parameter considered in the present study was turbidity. This parameter scores the presence of solid particles in suspension [43]. However, although the parameters turbidity and total solids are associated with each other, they are not absolutely equivalent [44]. In the analysis performed with the studied effluents, both presented high turbidity values (369 NTU and 173 NTU for DE and CE, respectively) for Class 2 rivers, compared to the recommended value of ≤100 NTU (Resolution No. 357/2005). High turbidity values tend to compromise light scattering, which affects the process of photosynthesis and, consequently, the functioning of aquatic ecosystems [45].

Nkansah et al. [46] attribute the high turbidity of salon effluents to the cosmetics components, such as volatile organic compounds, methacrylates, phthalates, sulfates, parabens, and formaldehyde. The authors also highlight that the turbidity derived specifically from hair dyeing processes is related to the excess of emulsifiers, composed of precursor agents, couplers, and oxidants. The altered turbidity results of the DE and CE corroborate the pH and conductivity data, thus reaffirming the polluting potential of hair dyes for aquatic environments.

According to Article 18 of the V Decree No. 8468/1976 [47], effluents from any polluting source can only be released into water resources, directly or indirectly, if they present dissolved oxygen (DO) concentration greater than 5.0 mg/L. The DO value of CE (5.60 mg/L) was very close to the reference value mentioned above, whereas DE presented a value considerably lower (2.37 mg/L) than recommended by Brazilian law. This low concentration of DO is worrisome because this parameter is a limiting factor for the maintenance of aquatic life, as well as for the processes of self-purification of natural aquatic systems [48,49,50]. Oxygen is necessary for aerobic organisms [36], so its depletion impairs the physiological processes and metabolic rate of aquatic species [51]. These data corroborate the records for pH, conductivity, and turbidity, confirming, once again, the concern of releasing these effluents directly into water bodies.

The effluents analyzed presented characteristics that can compromise the receiving water body, especially if released without treatment and in low-flow rivers. However, as the composition of salon effluents may vary according to the hair coloring process and the use of additional products, traditional physicochemical analyses serve only to identify and quantify the present compounds, but not to evaluate the biological effects and their potential risk [52,53]. Thus, physicochemical analyses are important as auxiliary information to ecotoxicity tests.

Aquatic ecotoxicity analyses should be done in a multifaceted way, with bioindicators of distinct taxonomic groups and different trophic levels [23]. The analyses should also consider the variability of the organisms’ sensitivity and the possible impacts on the ecosystem. The species *Daphnia similis*, *Artemia salina*, and *Danio rerio* have been successfully applied in this type of evaluation, so they were used as bioindicators of this study.

The species *D. similis* is a freshwater microcrustacean (order Cladocera), also known as a water flea. It is a species widely used in ecotoxicity tests because of its place in the aquatic trophic chain as an important food source for fish [54,55,56]. The organism has a relatively short life cycle, is easy to grow in the laboratory, and, due to its small size, requires low volumes of samples to perform the tests. In addition, it is an organism that presents high sensitivity to several aquatic contaminants [57,58,59].

The assays developed with *D. similis* showed a high sensitivity to both samples (DE EC_50_ = 0.54%; CE EC_50_ = 3.43%), with DE being more toxic than CE for this species. Associating these EC_50_ results with the physicochemical profile, it is possible to link the higher toxicity of DE with the higher turbidity (369 NTU), higher conductivity (391.0 μS/cm), and lower DO concentration (2.37 mg/L) of this effluent. These results corroborate the data of Melo et al. [53,60], who also reported higher acute toxicity to *D. similis* when exposed to effluents from the hair products industry, whose turbidity ranged from 313 to 5000 NTU.

Another microcrustacean used in the ecotoxicity evaluations of this study was *A. salina.* This saltwater microcrustacean is widely used as live food for small fish, which are the main representatives of secondary consumers in the food chains of marine environments [61,62]. This species of the order Anostraca is considered a good indicator of toxicity due to its short life cycle (2 to 4 months); high sensitivity to toxic substances; small size (8 to 12 mm for adults); high adaptability to various test conditions; low cost and rapid response test; and availability of commercial cysts that can be stored and used for long periods [63,64,65,66,67]. Based on the results presented here, this bioindicator showed a higher ecotoxicity for DE (LC_50_ = 3.874%) when compared to CE (LC_50_ = 8.327%). Although the results obtained in this evaluation with *A. salina* were, in general, similar to those of *D. similis*, the latter was more sensitive. Studies conducted in joint trials with *Daphnia* sp. and *Artemia* sp. by de Vega et al. [68] and Favilla et al. [69] also showed higher sensitivity in *Daphnia* sp. However, the direct comparison of the sensitivity of both species should be performed with great caution because there is a difference in the complexity of these test organisms.

Fish can also be used as bioindicators of toxicity tests. Among the most used fish species in ecotoxicology is *D. rerio*, popularly known as zebrafish [70,71,72]. Zebrafish have characteristics that make them an excellent vertebrate animal model for toxicity assessment, such as small size; easy maintenance; high reproductive rate; easily observable and quantifiable behavior; and having about 70% genetic homology with mammals [73,74,75].

The bioassays with *D. rerio* performed in this study showed higher toxicity to CE compared to DE. We also observed an increase in toxicity after 144 h of exposure to both effluents (DE: 24 hpf − LC_50_ = 6.59%; 96 hpf − LC_50_ = 6.59%; 144 hpf − LC_50_ = 6.37% and CE: 24 hpf − LC_50_ = 4.93%; 96 hpf − LC_50_ = 3.90%; 144 hpf − LC_50_ = 3.90%). Generally, TiO_2_-NP is used in cosmetic products, mainly in the production of pigments [76,77], so the toxicity observed in this assay may be associated with the presence of these compounds in the studied hair dye (see the cosmetics ingredients in Appendix A). There is now a growing concern about the harmful effects of these nanoparticles, due to their ability to associate with colloids naturally present in the aquatic environment [77].

In this context, the greater toxicity presented by the CE can be attributed to the association of TiO_2_-NP with the organic matter of this effluent, since it presents, in addition to the components of the dye, ingredients of the shampoo and conditioner. Thus, it is possible to infer that the association of TiO_2_-NP with the components from shampoo and conditioner increases the toxicological potential of CE. According to Vale et al. [78], TiO_2_-NP can interact with molecules of the biotic and abiotic environment of natural systems, generating effects that remain unknown. Adam et al. [79] warn that the reactivity of TiO_2_-NP with other elements is one of the most important processes to understand and predict how these particles are transported in the environment and what could be their potential effects on organisms. According to the study developed by Zhu et al. [62], exposure to TiO_2_-NP resulted in toxic effects on embryos and larvae of *D. rerio*, which corroborates the information and discussions of the present study.

Egg mortality was another endpoint evaluated, and the egg coagulation process was evident after 144 hpf of exposure to DE (12.5%) and CE (3.125 and 6.25%). Some studies reinforce that the toxicity of dyes may be a result of the action of TiO_2_-NP. According to Chen [80] and Clemente et al. [81], TiO_2_-NP tend to adhere to the chorion of the egg, forming a white and black outer layer. This event was also observed in this study (Figure 1c), more intensely for CE exposure. This uncommon structure may have caused the coagulation of the eggs and the embryonic development disruption, observed in the highest concentrations of both effluents.

There is evidence that TiO_2_-NP can adhere to the chorion of the embryo, being absorbed and evenly distributed across the tissues of the fish, without any tissue specificity [82]. Also, some studies cite that TiO_2_-NP toxicity would be caused by dyspnea and hypoxia, resulting from their adsorption to the surfaces of respiratory organs and subsequent tissue damage [83,84].

The increase in toxicity, which occurred in a time-dependent manner for both effluents, can also be attributed to the interaction of TiO_2_-NP and chorion. Ma and Diamond [85] suggest that this interaction may interfere with the oxygen transport process and that waste products may generate reactive oxygen species (ROS). Among the possible changes in redox balance, TiO_2_-NP may accelerate the production of hydroxyl radical (·OH), through the reactions of Fenton and Haber–Weiss, because it is based on a transition metal. TiO_2_-NP can also dissociate into cations, promoting competition with enzymatic factors for the allosteric site of superoxide dismutase, causing its inhibition [86,87]. Thus, it is estimated that the production of ROS, over the exposure time, may cause an oxidative disturbance and, consequently, potentiate the toxicity of both DE and CE.

In addition, edema of the pericardium and yolk sac was observed after 72 hpf when exposed to 6.25% of DE. According to the literature, edema is characterized as an abnormal accumulation of fluid in any tissue of the body [88]. The appearance of pericardial edema may be associated with cardiac dysfunction, acting as an indicator of osmotic or metabolic dysfunction, which is often correlated with the extravasation of endothelial vessels [89,90].

Verma et al. [91] found morphological and anatomical changes in zebrafish when the animals were treated with lower (50 µg mL^−1^) and higher (250 μg mL^−1^) concentrations of TiO_2_. In addition, the authors also observed the presence of abnormalities, such as deformation of the chorion and yolk sac at 48 hpf, whereas flexed tails, notochord malformation, and abnormal heart development were identified at 96 hpf. These effects were related to the accumulation of TiO_2_-NP in the different tissues of the animals [91]. In their review work, Roberto and Christofoletti [92] point out additional studies regarding the possible effects of TiO_2_-NP on chordate animals, such as *D. rerio, Pimephales promelas*, and *Xenopus laevis*.

Therefore, the endpoints adopted in the environmental toxicity tests presented in this work clearly demonstrate that the effluents containing hair dyes induced significant losses in the three test organisms. Although the effects varied among the bioindicators, the effluents containing capillary dyes acted on multiple targets, which suggests that they could compromise the population dynamics of aquatic ecosystems.

## 5. Conclusions

Considering all of the results reported here, we can conclude that the effluents generated in beauty salons after the use of hair dye (in this case, the brown color), associated or not with shampoo and conditioner, present high toxic potential concerning the aquatic biota as they induced deleterious effects to all studied organisms (*Daphnia similis*, *Artemia salina*, and *Danio rerio*).

Among the species used in the present study, *D. similis* was the one with the highest sensitivity, followed by *A. salina* and *D. rerio*. Although it was possible to determine and differentiate the toxic sensitivities of each indicator, the tests performed with *D. rerio* allowed a slightly better understanding of the effects of hair dye effluents. However, additional studies are still needed to elucidate the possible mechanisms of action of these compounds on each of the organisms tested.

The ecotoxicological evaluation proposed here was developed through tests performed with organisms of different taxonomic groups and trophic levels, to simulate, in a simplified way, the impacts that hairdressing salon effluents promote in aquatic ecosystems when released into the environment. Our results are corroborated by other toxicity studies performed with hair dyes, so it can be concluded that this class of effluent must receive specific prior treatment before being discarded in the environment. This study also provides support and highlights the need to create water ethics policies, especially concerning emerging contaminants from cosmetics, whose disposal is not yet regulated.

## Figures and Tables

**Figure 1 toxics-11-00911-f001:**
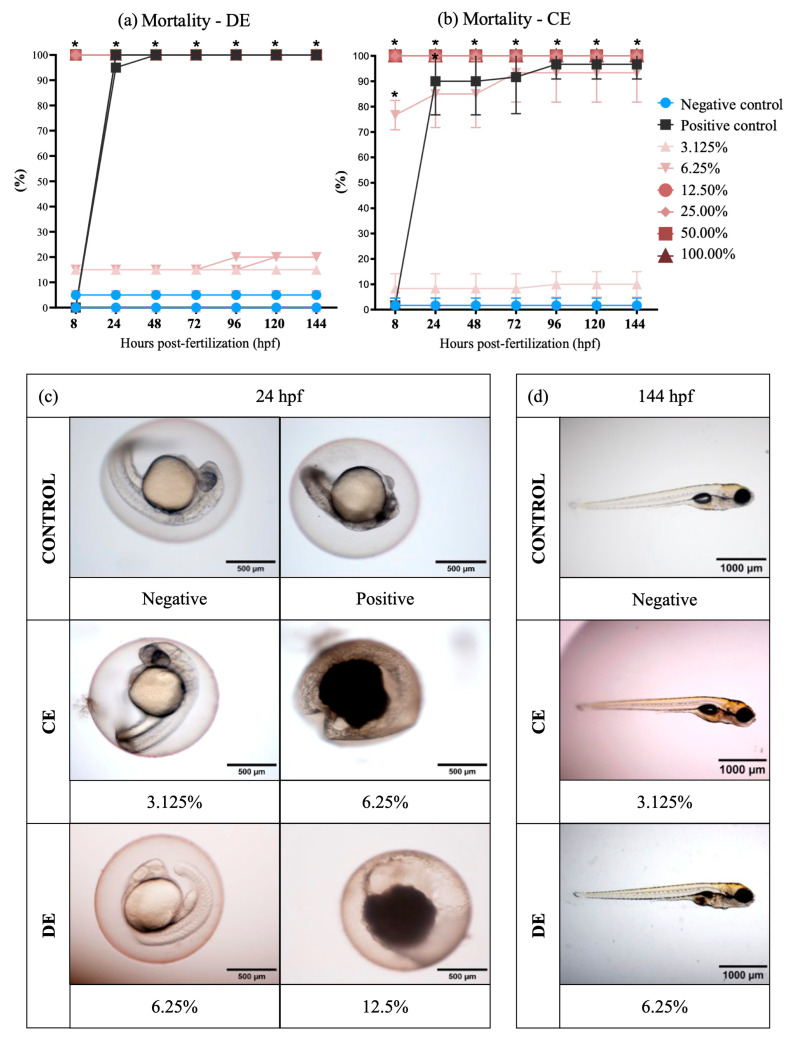
Mortality in zebrafish (*Danio rerio*) embryos. (**a**,**b**) Graphic representation of zebrafish embryo and larvae mortality, expressed by the proportion of coagulated eggs and absence of heartbeats per effluent, concentration, and exposure time. The negative control corresponds to the healthy embryos not exposed to effluents. Significant mortality of exposed zebrafish (*p* ≤ 0.05) relative to the control is identified with (*). Prepared with GraphPad Prism 5.01 software. (**c**) Micrographs showing zebrafish mortality: at the 6.125% dilution compared to negative control and the 3.125% dilution in the associated effluent (CE); and at the 12.5% dilution compared to the negative control and the 6.25% dilution in the non-associated effluent (DE) (2× magnification on MVX10 microscope, Olympus, Tokyo, Japan). Photographs are representative (*n* = 3) and were taken at 24 hpf. (**d**) Micrographs showing zebrafish survival from dilutions of 3.125% in the associated effluent (CE) and 6.25% in the non-associated effluent (DE) at the end of the exposure (1.6× magnification on Olympus MVX10 microscope). Photographs are representative (*n* = 3) and were taken at 144 hpf.

**Figure 2 toxics-11-00911-f002:**
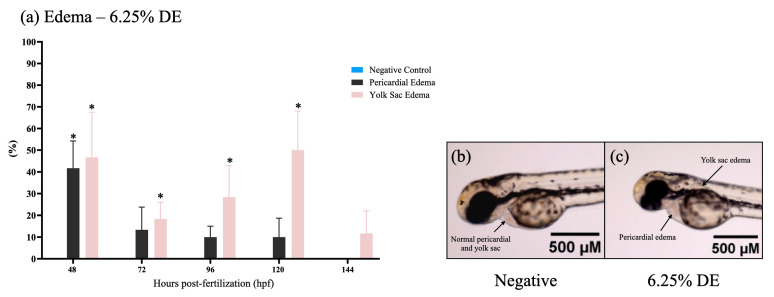
Pericardial and yolk sac edema in zebrafish (*Danio rerio*). (**a**) Graphic representation of pericardial edema rate and yolk sac edema rate in zebrafish larvae expressed by the incidence of this effect by DE dilution 6.25%, and exposure time. The negative control corresponds to the healthy embryos not exposed to effluents. The incidence of exposed zebrafish was significant if (*p* ≤ 0.05) relative to the control, as identified with (*). Prepared with GraphPad Prism 5.01 software. (**b**) Micrograph of the absence of pericardial edema and yolk sac edema in the negative control. (**c**) Micrograph showing the presence of pericardial edema and yolk sac edema in zebrafish exposed to DE (non-associated) effluent at 6.25% dilution (1.6× magnification on Olympus MVX10 microscope). Photographs are representative (*n* = 3) and were taken at 72 hpf.

**Figure 3 toxics-11-00911-f003:**
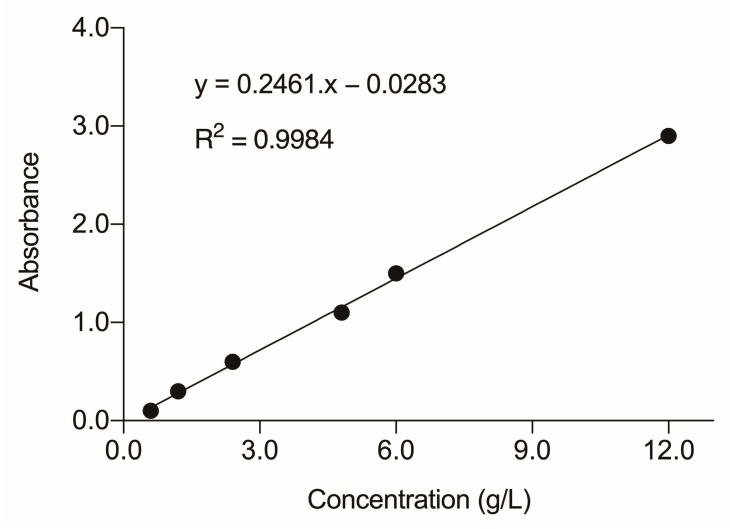
Standard curve for hair dye mixture obtained by linear regression. Readings were performed at 290 nm, the most suitable wavelength for maximum light absorption.

**Table 1 toxics-11-00911-t001:** Mean mortality of *Artemia salina* during the 48-h exposure to different concentrations of DE and CE.

Treatments	DE	CE
NC	0.00%	0.00%
3.125%	22.50%	6.25%
6.250%	33.75%	25.00%
12.500%	41.25%	28.75%
25.000%	100.00%	100.00%
50.000%	100.00%	100.00%
100.000%	100.00%	100.00%
LC_50_	3.874%	8.327%

NC: negative control; DE: dye effluent; CE: complete effluent (hair dye, shampoo, and conditioner); LC_50_: average lethal concentration.

**Table 2 toxics-11-00911-t002:** Mean mortality of *Daphnia similis* for 48 h exposure to different concentrations of DE and CE.

Treatments	DE	CE
NC	0.00%	0.00%
0.50%	45.00%	-
1.00%	90.00%	0.00%
3.00%	100.00%	15.00%
5.00%	-	100.00%
3.125%	100.00%	100.00%
6.25%	100.00%	100.00%
12.50%	100.00%	100.00%
25.00%	100.00%	100.00%
50.00%	100.00%	100.00%
100.00%	100.00%	100.00%
EC_50_	0.54%	3.43%

NC: negative control; DE: dye effluent; CE: complete effluent (hair dye, shampoo, and conditioner); EC_50_: average effective concentration.

**Table 3 toxics-11-00911-t003:** Results of LC_50_ after exposure to DE and CE during different periods (24, 96, and 144 hpf).

Stage Kimmel et al. [31]	Exposure Period (hpf)	DE		CE	
		Test Range (%)	LC_50_ (%)	Test Range (%)	LC_50_ (%)
Pharyngula	24	0.5–12.5	7.33 (6.25–8.65)	3.125–12.5	4.59 (3.94–5.36)
Early larva	96	0.5–12.5	7.73 (6.70–8.97)	3.125–12.5	4.25 (3.67–4.93)
Early larva	144	0.5–12.5	8.18 (7.13–9.40)	3.125–12.5	4.25 (3.67–4.93)

Hpf: Hours post-fertilization; DE: Dye effluent; CE: Complete effluent.

**Table 4 toxics-11-00911-t004:** Results of physicochemical analyses of DE and CE samples, obtained with the multiparameter probe.

Parameters	Samples		Values Recommended by Brazilian and International Legislation		
			CONAMA n° 357/2005	CONAMA n° 430/2011	Directive 91/271/EEC of 21/05/1991
	DE	CE			
Temperature (°C)	17.89	19.46	<40.0	<40.0	---
pH	6.16	5.14	5.00 to 9.00	5.00 to 9.00	---
Conductivity (µS/cm)	391.0	204.0	---	---	---
Dissolved oxygen (mg/L)	2.37	5.60	≥5.00	---	---
TSS (g/L)	0.254	0.126	≤0.5	---	35–60
Turbidity (NTU)	369	173	≤100	---	---
Salinity	0.01%	0.00%	≤0.5%	---	---

DE: Dye effluent; CE: Complete effluent; TSS: Total soluble solids; EEC: European Economic Community.

## Data Availability

The data presented in this study are available upon request from the corresponding authors.

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
