# Peer review of "Toxicity of Beauty Salon Effluents Contaminated with Hair Dye on Aquatic Organisms"

_toxics, 2023, doi:10.3390/toxics11110911_

Round 1

Reviewer 1 Report

Review of the manuscript Jurnal/ Journal Code für Paper

Toxicity of beauty salon effluents contaminated with hair dye on aquatic organisms

Authors: Gonçalves et al.

Dear authors of the above manuscript: below I try to answer the questions to increase the readability of the review

General remarks

What is the main point of the paper?

The analysis of waste water from hair / beauty salons with three ecotox tests

Title

Does the title represent the aims and conclusions? yes

Is the title accurate? yes

Content

Are the aims clear, and does the research address the aims? yes

Does the writing stick to that point? Yes and no

Is the manuscript accurate? yes

Is the manuscript concise? Yes, but sometimes it seems that something more in deep information about the effluent constitutents is missing.

Abstract

Mistakes in writing: -

General comments: the usage of the abbreviations CE and DE in the abstract is irritating as one wonders if CE comes from conditioner (the word next to it) or what DE means. Both are first introduced in the Material and Methods part. The authors should explain it much earlier or rewrite the abstract so that the abbreviations CE and DE are not needed any more.

Keywords: as the literature search engines always (at minimum) search title, abstract and keywords you can add in the keyword part all keywords which were not already in title and abstract!

The keyword „aquatic biomarker“ does not seem to fit to well as biomarkers are usually defined more like physiological or biochemical changes in the organisms.

Introduction

Mistakes in writing: in the fourth paragraph the word „dissolved“ is used twice

in the sixth paragraph: „… , especially for aquatics.“… sounds a little weird to me.

Comments: the first and second citations cite papers about more specifically fungis, diatoms and protists. From my point of view these citatitions are either not needed (as this is common knowledge that the aquatic like any other environment is complex etc.) or the authors should add other organism groups.. But I would prefer to leave these two citations out as they don‘t add knowledge which is specifically needed for the introduction. As with the citations of 7& 8. They both are educational books – so it would be common knowledge not needed to be strengthen by this citations. Citations #11 is about an insecticide, and a fungicide. I am not sure how this fits to the argumentation in the introduction. The authors should explain that- unless these substances are known to be part of hair / dying products. Overall the first 6-8 paragraphs of the introduction are a bit lenghty and do not introduced to well into the problem of beauty salon effluents.

Methods

The methods are well written but still some things need to added to be fully copied/ used/ redone by others:

comment:

what is meant by 0.5 and 5% treatments in paragraph 2.2 used for test validation ? I don‘t the „test validation“ part- what is meant here?

Paragraph 2.3.1. : what company was the BOD incubator from? (BOD – biological oxygen demand?) and what does exactly „reconstituted water“ mean ? (as it is not explained elsewhere- unless I did oversaw it)

Paragraph 2.3.3.2: how does the age and testing of fish embryos older than 120 confer with the animal testing law? I thought animal testing as in the OECD guideline 236 talks about a testing up to the age of 96 hpf? In some countries it is acknowledged that the age of 120 hpf is still outside of the animal testing range (i.e. start of independent feeding etc.) but 144hpf seems to be longer than usual. The authors should clearly state what they got from the longer exposure and that it is within the brazilian animal testing law.

Paragraph 2.5: please give the „g“ i.e. centrifugal force and not the „rpm“ as that is not giving the information for centrifuging as the rpm might be similar but on different centrifuges the g will change due to the size of the rotor etc.

Results

Are the contents of the manuscript relevant/ interesting/new?

Relevant: yes

Interesting: yes, but sometimes hard to read

Are the tables and the references relevant, or are there too many? There are too many tables (table 1 & 2 could better shown as figures

the tables would be easier to read if the authors would produce dose-response curves. With that they could even combine all results of all three test organisms and the two effluents in one figure. Thus a comparison and discussion would be easier.

If the authors stick to the tables – then change „Tratamentos“ in Table 2 to „treatments“ like in table 1. But again: I think it would help more in seeing differences in sensititivity between the three organisms if they would be combined in one figure with the x-axis as the concentration in % and the y -axis with the effects 0- 100%.

Table 3: Lthe LC50 of the embryos in pharyngula (24hpf) and early larvae (96hpf) are similar (6.59 %) but in the text in paragraph 3.3 it is written 7.33 %?

Table 4: the pH of the CE effluent is pretty low, and the oxygen of CE sample is 2.37: how does this may explain the toxic effects observed?

Discussion: comment: quite a lot of brazilian citations. Not used to that, but if it is o.k. with the other reviewers I would not mind. It occured to me that these citations might make a lot of sense if this hair dye problem would be very specific for brazil. But I think that is a more general problem and the authors might think to add more citations from other countries.

Language:

Understandability of the writing? Yes, but with some hard parts in the discussion.

Is the writing clear and the tone appropiate? yes

How is grammar, spelling and style? Fine to me (but I am a non-native speaker...)

Style of writing:

Long sentences? no

Usage of precise vocabularies? Most often

Multiple use of the same wording? no

Organization of the paper: fine

Did the authors arouse interest in the reader?

Yes, but I was lost at some parts as the authors did not go into much detail about the real constituents of the effluents. It kept a little on the surface, which is a pity as the idea of the whole paper is nice and important.

Recommendation:

Accept after minor revisions (minor as the changes do not include major additional experiments but just text work).

If the above points are attended to I would have no hesitation in recommending this paper for publication in Toxics

Author Response

We appreciate all your comments and suggestions, and we understand the intent to improve the article's clarity and therefore its quality. In the manuscript file, we have made some adjustments to the authors' filiation - English translation of their names - and English revision. Below, we pointed out some explanations, while the manuscript file has the corrections and the changes highlighted in red colour (please, see the new version of the manuscript). All italic text was already presented in the review report.

Reviewer #1:

Comments and Suggestions for Authors

Review of the manuscript Jurnal/ Journal Code für Paper: "Toxicity of beauty salon effluents contaminated with hair dye on aquatic organisms"

Authors: Gonçalves et al.

Dear authors of the above manuscript: below I try to answer the questions to increase the readability of the review.

Authors' answer: We really appreciate your comments and suggestions on our manuscript. It is possible to see a clear effort to improve the clarity of the article. Thank you.

Title:

Does the title represent the aims and conclusions? yes

Is the title accurate? yes

Authors' answer: No action is requested.

Content:

Are the aims clear, and does the research address the aims? yes

Does the writing stick to that point? Yes and no

Is the manuscript accurate? yes

Is the manuscript concise? Yes, but sometimes it seems that something more in deep information about the effluent constituents is missing.

Authors' answer: As we did not perform specific chemical analyses, which were not accessible to us, like chromatographic methods, we lacked information to go deep into the discussion. Besides that, we are providing the chemical composition of the cosmetic as supplementary material (see the attached file).

Components of brown hair dye, shampoo and hair conditioner.

PRODUCTS

DYE

HYDROGEN PEROXIDE SOLUTION

SHAMPOO AND CONDITIONER

Water

Water

Water

Cetostearyl alcohol

Hydrogen peroxide

Sodium Lauryl Ether Sulphate

Ethanolamine

Cetostearyl alcohol

Vegetable ingredient derived from coconut oil

Laurylether

Ceteareth-25

Cocamidopropyl betaine

Sodium Lauryl Ether Sulphate

Salicylic acid

Disterated cocoa

Glycerol Monostearate

Phosphoric acid

Fragrance

2,5-Diaminotoluene

Disodium phosphate

Dmdm Hydantoin

Sodium sulfate

Etidronic acid

Polyquaternium - 7

Myristyl alcohol                                                  

Dodium chloride

Sodium lauryl sulfate

Citric acid

Beeswax

Disodium EDTA

Sodium Cocoyl Isothionate

Sodium citrate

Perfume

Thickener

Shimmering / pearlescent pigment

Benzoic acid

Hydroxyethyl starch - 3,4 - methylmedioxyanililine HCl

Hydrolyzed silk

Resorcinol 

Phenoxyethanol

Sodium sulfite

Cotton

M- Aminophenol

Shea Butter

Ascorbic acid

Black tea

EDTA- Disodium Phosphate

Green tea extract

Hydrolyzed keratin

Chamomile

2- methylresorcinol

Shea

Titanium dioxide (CI 77891)

Cinnamon

Hexyl cinnamaldehyde

Coconut extract

Limonene

Commiphora myrrha (African Myrrh) resin extract

Benzyl Benzoate

Macadamia oil

Phenylenediamines (Diaminotoluenes)

Olive oil/olive oil

Abstract

Mistakes in writing: -

General comments: the usage of the abbreviations CE and DE in the abstract is irritating as one wonders if CE comes from conditioner (the word next to it) or what DE means. Both are first introduced in the Material and Methods part. The authors should explain it much earlier or rewrite the abstract so that the abbreviations CE and DE are not needed any more.

Authors' answer: We've added the explanation in the abstract: (complete effluent - CE) / (dye effluent - CE). That would be the best choice because we have used these abbreviations to show the results.

Keywords

As the literature search engines always (at minimum) search title, abstract and keywords you can add in the keyword part all keywords which were not already in title and abstract!

The keyword "aquatic biomarker" does not seem to fit to well as biomarkers are usually defined more like physiological or biochemical changes in the organisms.

Authors' answer: We agree with the reviewer, so we replaced "aquatic biomarker" with "ecotoxicity".

Introduction

Mistakes in writing: in the fourth paragraph the word "dissolved" is used twice in the sixth paragraph: "… , especially for aquatics.“… sounds a little weird to me.

Authors' answer: We are sorry about this lapse, so we have removed the duplicate word. About the other sentence, we highlight the aquatic organisms, as they are much more dependent on water and have greater exposure and interaction with this environmental matrix. Considering that, we added a reference to support this concept - (Wang, 2011).

Reference:

Wang W-X. Incorporating exposure into aquatic toxicological studies: An imperative. Aquatic Toxicology, 105S:9-15, 2011. doi:https://doi.org/10.1016/j.aquatox.2011.05.016.

Comments: the first and second citations cite papers about more specifically fungis, diatoms and protists. From my point of view these citatitions are either not needed (as this is common knowledge that the aquatic like any other environment is complex etc.) or the authors should add other organism groups.. But I would prefer to leave these two citations out as they don‘t add knowledge which is specifically needed for the introduction. As with the citations of 7& 8. They both are educational books – so it would be common knowledge not needed to be strengthen by this citations. Citations #11 is about an insecticide, and a fungicide. I am not sure how this fits to the argumentation in the introduction. The authors should explain that- unless these substances are known to be part of hair / dying products. Overall the first 6-8 paragraphs of the introduction are a bit lenghty and do not introduced to well into the problem of beauty salon effluents.

Authors' answer: We decided to revise citations 1 and 2, and remove citations 7 and 8. The change made in the first paragraph aims to bring more depth to literature citations, showing the relevance of the study to multiple disciplines. Regarding the other part, we agreed that is common knowledge and was also prolonging the introduction section, explaining its removal. As other reviewers suggested to shorten the introduction, we can attend to both suggestions, improving the quality of the article. We also changed citation 11 to another reference that discusses the inefficient removal of emerging contaminants by wastewater treatment plants. At last, we revised the introduction in order to better contextualize the beauty salon effluents issue.

References:

Zhou Y, et al. Current research trends on cosmetic microplastic pollution and its impacts on the ecosystem: A review. Environmental Pollution, 121106, 2023.

Chowdhury A, Naz A, Maiti SK. Distribution, speciation, and bioaccumulation of potentially toxic elements in the grey mangroves at Indian Sundarbans, in relation to vessel movements. Marine Environmental Research, 106042, 2023.

Morin-Crini N, Lichtfouse E, Liu G, et al. Worldwide cases of water pollution by emerging contaminants: a review. Environmental Chemistry Letters, 20:2311–2338, 2022. https://doi.org/10.1007/s10311-022-01447-4

Methods

The methods are well written but still some things need to added to be fully copied/ used/ redone by others.

Comments: What is meant by 0.5 and 5% treatments in paragraph 2.2 used for test validation? I don‘t the "test validation" part- what is meant here?

Authors' answer: Initially, the tests were performed with NC (0.0%) and several effluent concentrations (1.00; 3.00; 3.125; 6.25; 12.50; 25.00; 50.00; and 100.0%). However, as many concentrations above 3.00% induced lethality, the data distribution did not generate a reliable curve. So, considering that the concentrations of 0.5 and 5.0% were added to adjust the data distribution and the curve.

Paragraph 2.3.1. : what company was the BOD incubator from? (BOD – biological oxygen demand?) and what does exactly "reconstituted water" mean? (as it is not explained elsewhere- unless I did oversaw it)

Authors' answer: BOD incubator is from biological oxygen demand. In order to make the article clearer, we have added this explanation to the text. Regarding the 2.3.1. item, reconstituted water is the expression present in the ABNT guideline that means synthetic seawater. It is explained in the last sentence of the second paragraph. We have highlighted this part with a yellow background to ease the reviewer's recognition.

Paragraph 2.3.3.2: how does the age and testing of fish embryos older than 120 confer with the animal testing law? I thought animal testing as in the OECD guideline 236 talks about a testing up to the age of 96 hpf? In some countries it is acknowledged that the age of 120 hpf is still outside of the animal testing range (i.e. start of independent feeding etc.) but 144hpf seems to be longer than usual. The authors should clearly state what they got from the longer exposure and that it is within the brazilian animal testing law.

Authors' answer: We have followed OECD 236 guideline with some adaptations. We decided to extend the exposure time so that we could observe the organs for a longer period of time, mainly the swim bladder, which is of great importance as it is responsible for maintaining the balance of the fish in a certain position underwater. Other recent studies also have been used this time of exposure for zebrafish (Danio rerio) embryos (Cao et al., 2023; Chen et al., 2023; Santos et al., 2023).  Here in Brazil, all procedures with zebrafish must be approved by the Ethics Committee on Animal Use (CEUA), and this study was approved and registered under n°1392/2021.

References:

Cao X, Fu M, Du Q, Chang Z. Developmental toxicity of black phosphorus quantum dots in zebrafish (Danio rerio) embryos. Chemosphere. 2023 Sep;335:139029. doi: 10.1016/j.chemosphere.2023.139029. Epub 2023 May 25. PMID: 37244547.

Chen C, Zuo Y, Hu H, Li X, Zhang L, Yang D, Liu F, Liao X, Xiong G, Cao Z, Zhong Z, Bi Y, Lu H, Chen J. Hepatic lipid metabolism disorders and immunotoxicity induced by cysteamine in early developmental stages of zebrafish. Toxicology. 2023 Jul;493:153555. doi: 10.1016/j.tox.2023.153555. Epub 2023 May 24. PMID: 37236339.

Santos C, Valentim AM, Félix L, Balça-Silva J, Pinto ML. Longitudinal effects of ketamine on cell proliferation and death in the CNS of zebrafish. Neurotoxicology. 2023 Jul;97:78-88. doi: 10.1016/j.neuro.2023.05.008. Epub 2023 May 15. PMID: 37196828.

Paragraph 2.5: please give the "g" i.e. centrifugal force and not the "rpm" as that is not giving the information for centrifuging as the rpm might be similar but on different centrifuges the g will change due to the size of the rotor etc.

Authors' answer: We have used the Clinical Laboratory Centrifuge 80-2B Analogical - 12 x 15 mL - max.: 4,000 rpm, brand: Centribio (Brazil). Then, based on a rotor of 48 mm of radius, we will have ~484 × g at 3,000 rpm. We have altered this information in the article.

Results

Are the contents of the manuscript relevant/ interesting/new?

Relevant: yes

Interesting: yes, but sometimes hard to read

Are the tables and the references relevant, or are there too many? There are too many tables (table 1 & 2) could better shown as figures the tables would be easier to read if the authors would produce dose-response curves. With that they could even combine all results of all three test organisms and the two effluents in one figure. Thus a comparison and discussion would be easier.

If the authors stick to the tables – then change "Tratamentos" in Table 2 to "treatments" like in table 1. But again: I think it would help more in seeing differences in sensititivity between the three organisms if they would be combined in one figure with the x-axis as the concentration in % and the y -axis with the effects 0- 100%.

Authors' answer: We created a figure that brought all Artemia salina and Daphnia similis results together. However, as we present the results in separate sections, we chose to keep the original layout, using the tables. Thus, the correction of Table 2 was done.

Table 3: Lthe LC50 of the embryos in pharyngula (24hpf) and early larvae (96hpf) are similar (6.59 %) but in the text in paragraph 3.3 it is written 7.33 %?

Authors' answer: We have revised all results presented in Table 3, in section 3.3, and in the abstract. Corrections were made.

Table 4: the pH of the CE effluent is pretty low, and the oxygen of CE sample is 2.37: how does this may explain the toxic effects observed?

Authors' answer: According to Osibanjo et al. (2011), pH is an important variable of water quality, as it influences several biological and chemical processes. Changes in the pH can affect the survival of aquatic organisms as their metabolic activities are dependent on it. Also, Omer (2019) explains that aquatic organisms are adapted to specific pH and changes can jeopardize them.

Dissolved oxygen (DO) is crucial to the survival of aquatic organisms and it represents the quality of a freshwater ecosystem (Osibanjo et al., 2011). Oxygen is necessary for aerobic organisms (Omer, 2019), so its depletion impairs the physiological processes and metabolic rate of aquatic species (Ferreira et al., 2008).

As we observed the lethality of test organisms, the information presented above can explain the effluent toxicity. We added some of this information to the text.

Reference:

Ferreira ALG, Loureiro S, Soares AMVM. Toxicity prediction of binary combinations of cadmium, carbendazim and low dissolved oxygen on Daphnia magna. Aquatic Toxicology, 89(1):28-39, 2008.

Omer NH. Water Quality Parameters. In: Summers JK. Water Quality - Science, Assessments and Policy. IntechOpen: 2019. http://dx.doi.org/10.5772/intechopen.89657

Osibanjo O, Daso AP, Gbadebo AM. The impact of industries on surface water quality of River Ona and River Alaro in Oluyole Industrial Estate, Ibadan, Nigeria. African Journal of Biotechnology, 10(4):696–702, 2011.

Discussion

Comment: quite a lot of brazilian citations. Not used to that, but if it is o.k. with the other reviewers I would not mind. It occured to me that these citations might make a lot of sense if this hair dye problem would be very specific for brazil. But I think that is a more general problem and the authors might think to add more citations from other countries.

Authors' answer: We have approximately 70% of the citations based on international references, whereas Brazilian citations are most related to basic ecotoxicological knowledge or laws that regulate physicochemical parameters for effluents in Brazil. As the other reviewers did not mind, we believe the study has great international importance and the use of these references does not prejudice its transposition to similar scenarios in other countries.

Language:

Understandability of the writing? Yes, but with some hard parts in the discussion.

Is the writing clear and the tone appropiate? yes

How is grammar, spelling and style? Fine to me (but I am a non-native speaker...)

Style of writing:

Long sentences? no

Usage of precise vocabularies? Most often

Multiple use of the same wording? no

Organization of the paper: fine

Did the authors arouse interest in the reader?

Yes, but I was lost at some parts as the authors did not go into much detail about the real constituents of the effluents. It kept a little on the surface, which is a pity as the idea of the whole paper is nice and important.

Authors' answer: We understand that the real constituents of the effluents are important to its toxicity and we have explained our limitation above.

Recommendation:

Accept after minor revisions (minor as the changes do not include major additional experiments but just text work).

If the above points are attended to I would have no hesitation in recommending this paper for publication in Toxics.

Authors' answer: We really appreciate all your observations and suggestions. Hope we attend what the reviewer expected.

Reviewer 2 Report

It is my pleasure to go through the manuscript and suggest comments aimed at improvement of the article.

1. The connection between pollutant, its dispersal and bioaccumulation in biota need to be focused in initial paragraph of the introduction. Reders must get an understanding on why and how this study is relevent to multiple disciplines. Recent references,(from 2022 and 2023) can be used in this section. This put emphasis on the dpth of litereture review associated with the work and recent trend of research. There may be less work on cosmetics (which increases the novelity of this article, and need to be mentioned), but there are lot of recent research on different other form of pollutants in water-sediment-biota nexus, such as, https://www.sciencedirect.com/science/article/abs/pii/S0141113623001708, https://www.sciencedirect.com/science/article/abs/pii/S0269749123001082, https://www.sciencedirect.com/science/article/abs/pii/B9780323996846000021 etc.

2. Introduction part can be shortened. Just focus on recent research on pollutants-biota interaction (not limited to cosmetics), followed by mention on why studies on cosmetics is relevent, then mention the research gap and finish with a strong objective/hypothesis. All other general narative on importance of the study can be shifted to discussion section to build up relevence.

3. Authors may want o reframe the subsection title of 2.3.1. Any scientific names need to be italized. And the subsection need not. to be named after the model organism but on what is the main part of the section.

4. Same comment as 3 also applied to the section 2.3.2

5. Has the authors mentioned the compliance with 'research ethics board' as experiments on animal subjects has been used.

6. What are the quality control assuarence methods used in the study?

Overall, I find the article novel and new knowledge in the field of research. I will invite authors to respond to the comments.

English is understandable.

Author Response

We appreciate all your comments and suggestions, and we understand the intent to improve the article's clarity and therefore its quality. In the manuscript file, we have made some adjustments to the authors' filiation - English translation of their names. Below, we pointed out some explanations, while the manuscript file has the corrections and the changes highlighted in red colour (please, see the manuscript attached). All italic text was already presented in the review report.

Reviewer #2:

Comments and Suggestions for Authors

It is my pleasure to go through the manuscript and suggest comments aimed at improvement of the article.

Authors' answer: We really appreciate your comments and suggestions on our manuscript. It is possible to see a clear effort to improve the clarity of the article. Thank you.

  1. The connection between pollutant, its dispersal and bioaccumulation in biota need to be focused in initial paragraph of the introduction. Reders must get an understanding on why and how this study is relevent to multiple disciplines. Recent references,(from 2022 and 2023) can be used in this section. This put emphasis on the dpth of litereture review associated with the work and recent trend of research. There may be less work on cosmetics (which increases the novelity of this article, and need to be mentioned), but there are lot of recent research on different other form of pollutants in water-sediment-biota nexus, such as, https://www.sciencedirect.com/science/article/abs/pii/S0141113623001708 , https://www.sciencedirect.com/science/article/abs/pii/S0269749123001082 , https://www.sciencedirect.com/science/article/abs/pii/B9780323996846000021 etc.

Authors' answer: We have inserted some references to highlight the dispersal of pollutants and cosmetics in the environment. However, we tried to keep the text concise.

  1. Introduction part can be shortened. Just focus on recent research on pollutants-biota interaction (not limited to cosmetics), followed by mention on why studies on cosmetics is relevent, then mention the research gap and finish with a strong objective/hypothesis. All other general narative on importance of the study can be shifted to discussion section to build up relevence.

Authors' answer: We have removed a paragraph of the introduction section, but also added some other information. Minor modifications were performed to attend to all reviewers' suggestions.

  1. Authors may want o reframe the subsection title of 2.3.1. Any scientific names need to be italized. And the subsection need not. to be named after the model organism but on what is the main part of the section.

Authors' answer: We have changed the section name to "Test method with Artemia salina".

  1. Same comment as 3 also applied to the section 2.3.2

Authors' answer: We have changed the section name to "Test method with Daphnia similis".

  1. Has the authors mentioned the compliance with 'research ethics board' as experiments on animal subjects has been used.

Authors' answer: This information is presented in the section 2.3.3.1. - […] Experimental Research Unit of Botucatu Medical School, UNESP, Brazil, approved by CEUA n°1392/2021. Also, at the end of the manuscript, it was explained: Institutional Review Board Statement: The study was conducted in accordance with the Declaration of Helsinki, and approved by the Ethics Committee CEUA of UNESP, Brazil (protocol code n°1392/2021 of August 31, 2021).

Here in Brazil, all procedures with zebrafish must be approved by the Ethics Committee on Animal Use (CEUA), and this study was approved and registered under n°1392/2021.

  1. What are the quality control assuarence methods used in the study?

Authors' answer: All experiments were conducted based on scientific literature and well-established guidelines - ABNT NBR 16530/2016 for Artemia salina; ABNT NBR 12713/2016 for Daphnia similis; OECD No. 236/2013 for Danio rerio. Each of the guidelines presents which control must be done to ensure the quality of the results. The assays with A. salina were followed by a negative control based on the exposure of the animals to culture water (reconstitute water - artificial marine water). The assays with D. similis were carried out with a negative control based on the culture water. The assays with D. rerio also had negative control based on reconstituted water ISO-7346. All negative controls were applied to show basal damage that could happen to the test organisms and other groups were statistically compared to them.

Overall, I find the article novel and new knowledge in the field of research. I will invite authors to respond to the comments.

Authors' answer: We hope we have answered all questions adequately.

Comments on the Quality of English Language

English is understandable.

Round 2

Reviewer 2 Report

This article can be accepted in the current form.

English is good